# Cytomegalovirus detection is associated with ICU admission in non-AIDS and AIDS patients with *Pneumocystis jirovecii* pneumonia

Alexandre Perret[1,2], Marion Le Marechal[1,2], Raphaele Germi[2,3], Daniele Maubon[2,4], Cécile Garnaud[2,4], Johan Noble[5], Aude Boignard[6], Loïc Falque[7], Mathieu Meunier[8], Théophile Gerster[9], Olivier Epaulard[1,2,10]*

1 Infectious Disease Unit, Grenoble-Alpes University Hospital, Grenoble, France, 2 GRIC, CIC1408 INSERM-UGA-CHUGA, Bouliac, France, 3 Virology, Grenoble-Alpes University Hospital, Grenoble, France, 4 Mycology, Grenoble-Alpes University Hospital, Grenoble, France, 5 Nephrology, Grenoble-Alpes University Hospital, Grenoble, France, 6 Cardiology, Grenoble-Alpes University Hospital, Grenoble, France, 7 Pneumology, Grenoble-Alpes University Hospital, Grenoble, France, 8 Haematology, Grenoble-Alpes University Hospital, Grenoble, France, 9 Hepato-Gastro-Enterology, Grenoble-Alpes University Hospital, Grenoble, France, 10 IBS UMR 5075 CNRS-CEA-UGA, Grenoble, France

* oepaulard@chu-grenoble.fr

**Data Availability Statement:** The data can be find on the OSF depository, at the following link: https://

## Abstract

### Objectives

Cytomegalovirus (CMV) is frequently detected in lung and/or blood samples of patients with *Pneumocystis jirovecii* pneumonia (PJP), although this co-detection is not precisely understood. We aimed to determine whether PJP was more severe in case of CMV detection.

### Methods

We retrospectively included all patients with a diagnosis of PJP between 2009 and 2020 in our centre and with a measure of CMV viral load in blood and/or bronchoalveolar lavage (BAL). PJP severity was assessed by the requirement for intensive care unit (ICU) admission.

### Results

The median age of the 249 patients was 63 [IQR: 53–73] years. The main conditions were haematological malignancies (44.2%), solid organ transplantations (16.5%), and solid organ cancers (8.8%). Overall, 36.5% patients were admitted to ICU. CMV was detected in BAL in 57/227 patients; the 37 patients with viral load ≥3 log copies/mL were more frequently admitted to ICU (78.4% vs 28.4%, p<0.001). CMV was also detected in blood in 57/194 patients; the 48 patients with viral load ≥3 log copies/mL were more frequently admitted to ICU (68.7% vs 29.4%, p<0.001). ICU admission rate was found to increase with each log of BAL CMV viral load and each log of blood CMV viral load.

osf.io/3kesf/files/osfstorage/
6540f6be87852d1a56a5971c.

**Funding:** The authors received no specific funding for this work.

**Competing interests:** The authors have declared that no competing interests exist.

## Conclusions

PJP is more severe in the case of concomitant CMV detection. This may reflect either the deleterious role of CMV itself, which may require antiviral therapy, or the fact that patients with CMV reactivation are even more immunocompromised.

## Introduction

*Pneumocystis jirovecii* pneumonia (PJP), one of the predominant infections in AIDS, is now acknowledged as an important and increasing threat in other immunocompromised patients [1]. Even when treated, PJP is associated with a high mortality rate ranging from 31% to 60% for non-AIDS patients [2,3], and 10.3% to 13.5% for people with AIDS [4,5]. Several studies [4,6–8] (some of them conducted 30 years ago) reported the detection (by various means) of cytomegalovirus (CMV) in lung samples of AIDS patients with PJP (in a variable proportion depending on the studies). More recently, this has also been reported in non-AIDS patients [9–15], although it should be noted that the first description of this association existed before the AIDS pandemic was known in, for example, a patient receiving busulfan in 1974 [16].

The detection of CMV is still complex to integrate in the management of patients with PJP. When the CMV viral load reaches high values such as 10,000 copies/mL, antiviral therapy is usually introduced in addition to the anti-*Pneumocystis* treatment; the significance of lower values is not firmly established; e.g., no consensual attitude exist in case of a value of 1,000 copies/mL. Indeed, CMV detection may be interpreted as a non-pathogenic reactivation from the latency reservoir or as a *per se* deleterious co-infection, depending on the viral load; however, to date, no threshold values have been established.

The aforementioned studies provided contradictory results: some reported higher mortality and longer hospital or ICU stay in the case of concomitant CMV detection [8–11,13,14], while others reported no significant association [7,12]. Nevertheless, the number of studies on the subject remains limited. Further, the majority were performed more than 10 years ago, with a small number of subjects and rarely with non-AIDS patients.

We conducted a study to assess i) the frequency and intensity of CMV detection in non-AIDS and AIDS patients with *Pneumocystis* infection, and ii) the extent to which CMV detection was associated with severity of PJP.

## Materials and methods

### Objectives

Our main objective was to explore the association between outcome and CMV detection in patients with PJP. The primary endpoint was the need for intensive care unit (ICU) admission related to the PJP episode. We also assessed the length of ICU stay (if any) and 28-day mortality.

### Study design

We conducted a retrospective study on all adult (≥18 years) patients in Grenoble-Alpes University Hospital. We considered all patients with a diagnosis of PJP retained by clinicians between July 2009 and January 2020 (we did not retrospectively reassessed the diagnosis); PJP was usually retained in the case of an at-risk medical condition, with an evocative thorax CT scan and a positive *P. jirovecii* polymerase chain reaction (PCR) on bronchoalveolar lavage (BAL) or bronchial aspiration (the thresholds usually applied to fungal load are described

below). Among them, we included all patients in whom a search for CMV by PCR in the same BAL as *P. jirovecii*, or in blood in the week before or after the diagnosis of PJP, or both, had been performed.

## P. jirovecii and CMV viral load

In our institution, the diagnosis of PJP relies on the detection of P. jirovecii by PCR on BAL. This method was proven reliable [17]; its high sensitivity leads to use thresholds to differentiate infection from colonisation [18,19]. In the study period, two *P. jirovecii* PCRs amplifying the nuclear multicopy gene coding for the major surface glycoprotein were used in our institute: from 2009 to 2016, the quantitative *P. jirovecii* PCR was performed on the Light-cycler 2.0 (Roche Diagnostics, Mannheim, Germany), and from 2016 to 2020 this PCR was transferred identically to the BD-Max platform (Becton Dickinson, New Jersey, USA). For both PCRs, the interpretation thresholds were established as follows: detectable but < 3,000 copies/ml: probable colonisation; > 30,000 copies/ml: probable PJP; 3,000–30,000 copies/ml: possible PJP ("grey zone") [20].

For CMV viral load, nucleic acids were extracted from blood using automatic extraction technologies (MagNaPure LC® [Roche Diagnostics] from 2009 to 2017 and eMag [Biomérieux, Marcy l'Etoile, France] since 2017) and from BAL using EasyMag® (Biomérieux) automatic extraction technology. The CMV viral load was measured by real time qPCR by using a CMV R-gene® kit (Biomérieux) on a LightCycler® 480 Instrument II (Roche Diagnostics) according to the manufacturer's instructions. Results were expressed as copies/mL or log10 copies/mL. We conducted some on the analyses by separating patients with blood viral load above or under 3 log copies/mL; indeed, most of guidelines recommend not to treat infection under this value; conversely, the French guidelines for infection in AIDS patients propose to introduce antiviral for value above this threshold [21]. For BAL viral load, as BAL is an intrinsically diluted sample, the choice of the threshold is more complex; by analogy with blood, and even if the two compartments differ on many points, we used the same cut-off value of 3 log copies/mL.

## Data collection

Data for each patient were collected from their electronic medical files and anonymously analysed. Data were collected between the 1st of April and the 15th of May 2020. All patients had signed a consent form for the anonymous use of their medical data for research purposes (a decision that they could revoke at any moment), and information on the retrospective use of anonymised data for research purpose was available for all patients. Due to this process, in accordance with French legislation, it was not required to submit this specific project to an ethical board.

## Statistical analysis

The association between continuous and categorical variables was explored using the Mann-Whitney U test, the association between categorical variables using chi-square test, and the association between two continuous variables using the Spearman test.

Data (Excel table) are available on simple email request to the corresponding author.

## Results

### Study population

We included 249 patients (60.4% men, median age 63 years [IQR 53–73]) (Table 1). The immunosuppressive conditions were haematological malignancies (N = 110, 44.2%), solid

**Table 1. Patient characteristics according to intensive care unit (ICU) admission or not.**

| | | | Total N = 249 | Patients admitted to ICU N = 91 | Patients not admitted to ICU N = 158 | p |
|---|---|---|---|---|---|---|
| **Age** | | | 65 [53–73] | 66 [55–73] | 65 [53–73] | 0.679 |
| **Condition** | Haematological malignancies | | 110 | 34 | 76 | |
| | Solid organ transplantation | | 41 | 13 | 28 | |
| | HSCT | | 13 | 3 | 10 | |
| | Solid cancer | | 22 | 8 | 14 | |
| | Immunosuppressive therapy without transplantation or malignancies | | 36 | 15 | 21 | |
| | AIDS | | 15 | 10 | 5 | |
| | Other | | 12 | 8 | 4 | |
| **Blood lymphocyte count** | | | 0.40 [0.20–0.80] | 0.30 [0.10–0.52] | 0.40 [0.20–0.90] | **0.001** |
| **Gammaglobulin in blood** (g/L) | | | 6.7 [4.6–9.7] | 6.0 [3.8–8.4] | 7.4 [5.0–10.3] | **0.034** |
| *P. jirovecii* **load in BAL** Median [IQR] log copies/mL | | | 4.9 [4.0–5.9] | 5.0 [4.1–6.4] | 4.9 [4.0–5.7] | 0.166 |
| **BAL CMV viral load N = 227** | Median [IQR] log copies/mL | | 0 [0–1.76] | 0 [0–3.42] | 0 [0–0] | **<0.001** |
| | Undetectable | | 170 (74.9%) | 44 (53.0%) | 126 (87.5%) | **<0.001** (<3 log vs ≥3 log) |
| | Detectable <3 log | | 20 (8.8%) | 10 (12.0%) | 10 (6.9%) | |
| | 3 to 4 log | | 21 (9.2%) | 13 (15.7%) | 8 (5.6%) | |
| | ≥4 log | | 16 (7.0%) | 16 (19.3%) | 0 (0.0%) | |
| **Blood CMV viral load N = 194** | Median [IQR] log copies/mL | | 0 [0–2.94] | 0 [0–0.95] | 0 [0–0] | **<0.001** |
| | Undetectable | | 137 (70.6%) | 40 (52.6%) | 97 (82.2%) | **<0.001** (<3 log vs ≥3 log) |
| | Detectable <3 log | | 9 (4.6%) | 3 (3.9%) | 6 (5.1%) | |
| | 3 to 4 log | | 30 (15.5%) | 17 (22.4%) | 13 (11.0%) | |
| | ≥4 log | | 18 (9.3%) | 16 (21.1%) | 2 (1.7%) | |
| **Mortality at 28 days** | | | 30 | 24 (26.4%) | 6 (3.8%) | **<0.001** |

BAL: Bronchoalveolar lavage; CMV: Cytomegalovirus; HSCT: Haematopoietic stem cell transplantation.

Median *P. jirovecii* load in BAL was 4.9 log [4.0–5.9] copies/mL; the value was less than 30,000 in 95 patients, including 29 cases with values less than 3,000, meaning that the diagnosis of PJP also depended on other arguments in such cases. AIDS patients had the highest fungal load (7.2 [5.9–8.1] copies/mL).

organ transplantation (N = 41, 16.5%) (with 14 heart transplantations, 13 kidney transplantations, 8 liver transplantations, and 6 liver transplantations), haematopoietic stem cell transplantation (N = 13, 5.2%), solid cancer (N = 22, 8.8%), immunosuppressive therapy without transplantation or malignancies (N = 36, 14.5%), AIDS (N = 15, 6.0%), and others (N = 12, 5.2%). Median total blood lymphocytes (available in 244 patients at the time of PJP) was 0.4 G/L [0.2–0.8], and median total gammaglobulin levels (available in 145 patients at the diagnosis of PJP or in the preceding 3 months) were 6.7 g/L [4.7–9.7].

CMV PCR in BAL was available for 227 patients; CMV was detected in 57 patients (25.1%), including 21 with a viral load between 3 and 4 log copies/mL and 16 above 4 log copies/mL. CMV viral load and *P. jirovecii* load in BAL were correlated (p<0.001, R = 0.68), and patients with CMV load ≥ 3 log copies/mL had *P. jirovecii* load 10 times higher than those with a lower or undetectable CMV load (Table 2).

Blood CMV viral load was performed in 194 patients; CMV was detected in the blood of 57 patients (29.4%), including 30 with a value between 3 and 4 log copies/mL and 18 with a value above 4 log copies/mL. Those with blood CMV viral load ≥ 3 log copies/mL had *P. jirovecii* load 10 times higher than those with a lower or undetectable blood CMV load (Table 2).

BAL and blood CMV viral loads were correlated (p<0.001, R = 0.76): 113 patients had negative blood and BAL CMV viral loads, and 24 had blood and BAL CMV viral loads ≥3 log.

**Table 2. Patient characteristics according to cytomegalovirus (CMV) detection in bronchoalveolar lavage (BAL) or blood.**

| | BAL CMV viral load (N = 227) | | | Blood CMV viral load (N = 194) | | |
|---|---|---|---|---|---|---|
| | <3 log* (N = 190) | ≥3 log (N = 37) | p | <3 log* (N = 146) | ≥3 log (N = 48) | p |
| **Age (years)** Median [IQR] | 65 [53–73] | 66 [50–72] | 0.673 | 62 [52–73] | 67 [57–73] | 0.213 |
| **Blood lymphocyte count (G/L)** Median [IQR] | 0.40 [0.20–0.90] | 0.20 [0.10–0.42] | **0.003** | 0.40 [0.20–0.80] | 0.30 [0.10–0.65] | 0.185 |
| **Gammaglobulin in blood (g/L)** Median [IQR] | 6.7 [4.8–10.2] | 4.0 [2.8–7.5] | **0.002** | 6.6 [4.8–9.8] | 6.1 [3.7–8.8] | 0.312 |
| ***P. jirovecii* load in BAL (log copies/mL)** Median [IQR] | 4.8 [3.9–5.6] | 5.9 [4.8–8.1] | **<0.001** | 4.9 [3.9–5.6] | 5.9 [4.4–8.1] | **<0.001** |
| **ICU admission N (%)** | 54/190 (28.4%) | 29/37 (78.4%) | **<0.001** | 43/146 (29.4%) | 33/48 (68.7%) | **<0.001** |
| **ICU lenght of stay (days)** | 9 [4–18] | 18 [9–31] | **0.007** | 10 [4–14] | 19 [8–31] | **0.001** |
| **Mortality at 28 days N (%)** | 21/190 (11.0%) | 5/37 (13.5%) | 0.8814 | 17/146 (12.0%) | 4/48 (11.6%) | 0.709 |

ICU = intensive care unit; BAL = bronchoalveolar lavage.

* including patients with undetectable CMV viremia.

## ICU admission

Among the 249 patients, 36.5% (91/249) were admitted to ICU; 57 (22.9%) received orotracheal intubation (IOT). Age and *P. jirovecii* load did not differ according to ICU admission; nevertheless, blood lymphocyte count and gammaglobulin blood concentration were lower in the patients admitted to ICU (Table 1).

CMV viral load in BAL differed according to ICU admission: 34.9% of patients admitted to ICU had BAL viral loads higher than 3.0 log copies/mL versus 5.6% of those not admitted to ICU (p<0.001) (Table 1), while the median BAL CMV viral load was higher in the case of ICU admission (Fig 1). The same association was observed if only patients with *P. jirovecii* fungal

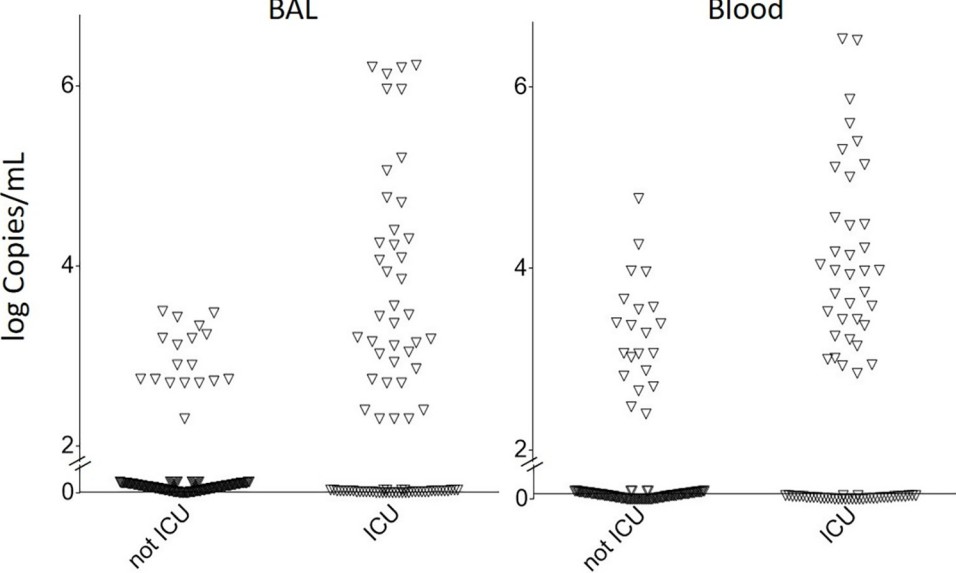

**Fig 1. Bronchoalveolar lavage (BAL) and blood CMV viral load in patients with or without intensive care unit (ICU) admission.**

load ≥30,000 were considered. The proportion of ICU admissions was 25.9%, 50.0%, 61.9% and 100% of patients with a BAL CMV viral load that was undetectable, 2 to 3 log, 3 to 4 log, or ≥4 log copies/mL, respectively (p<0.001). Moreover, patients admitted to ICU with a BAL CMV viral load ≥3 log copies/mL stayed in ICU twice as long as those with a BAL CMV viral load <3 log copies/mL or undetectable (Table 2).

Similarly, blood CMV viral load differed according to ICU admission: 43.5% of patients admitted to ICU had a blood viral load above 3 log copies/mL vs 12.7% of those not admitted to ICU (p<0.001) (Table 1), while the median blood CMV viral load was higher in the case of ICU admission (Fig 1). The same association was observed if only patients with *P. jirovecii* fungal load ≥30,000 were considered. The proportion of ICU admissions was 29.2%, 33.3%, 56.7% and 88.9% among those with a blood CMV viral load that was undetectable, 2 to 3 log, 3 to 4 log, or ≥4 log copies/mL, respectively (p<0.001). Moreover, patients admitted to ICU with a blood CMV viral load ≥3 log stayed in ICU twice as long as those with a blood CMV viral load <3 log copies/mL or undetectable (Table 2).

When considering both BAL and blood samples, 40 of the 61 (65.6%) patients with BAL and/or blood CMV viral load above 3 log copies/mL were admitted to ICU vs 51 of the 188 (27.1%) patients without BAL or blood viral load above 3 log copies/mL.

## Discussion

PJP incidence is increasing, mostly because more patients receive therapy that may lead to an immunocompromised state (especially anti-cancer chemotherapy and immunosuppression in transplantation and autoinflammatory diseases). Meanwhile, AIDS, once the most prominent disease associated with PJP, is less prevalent in most countries. AIDS-associated PJP and PJP associated with immunosuppressive therapy are sometimes considered two different diseases, with a higher fungal load [22] and lower fatality [23] in the former. Steroids have been shown to be favourable in AIDS-associated moderate-to-severe PJP [24], but the routine adjunctive use of glucocorticoids in non-HIV patients with PCP and respiratory failure is not recommended in non-HIV-infected haematology patients by the ECIL [25], or more broadly in non-HIV patients by some authors [26]; it is however recommended by the American Thoracic Society guidelines [27]. Interestingly, some authors warned against the use of steroids in PJP due to the risk of CMV infection in non-AIDS patients [8]. In both populations, concomitant CMV replication/detection has been reported, although the interpretation of this association is not consensual: as already stated, some studies reported a poorer outcome in the case of CMV detection [8–11,13,14], while others did not [7,12]. Recommendations have been extensively produced to guide CMV infection management in patients with AIDS and in other immunocompromised patients, although the specific situation of CMV replication in patients with PJP is not precisely addressed.

We observed that CMV detection in blood or BAL was associated with a worse outcome in PJP patients: those with detectable CMV DNA were more frequently admitted to ICU, while the proportion of ICU admissions increased with CMV viral load. Several interpretations may account for these findings.

Firstly, as CMV is a well-known agent of pneumonia in the immunocompromised host, CMV detection (particularly in BAL) may reflect CMV disease and not only CMV infection. Patients with detectable CMV DNA in BAL would therefore have more severe lung disease than those with only PJP. The fact that we observed a trend between the level of viral load (<3 log, 3 to 4 log, ≥4 log) and the proportion of ICU admission is an important argument for this interpretation. This is also in line with a study reporting that patients with mixed lung infections had higher levels of inflammatory cytokines in BAL than those with "pure" PJP [28]. In

addition, in a mouse model of CMV and *Pneumocystis* infection, more severe lung disease was observed in the case of co-infection [29]. Taken together, these data suggest that antiviral therapy should be applied in patients with PJP and a detectable CMV replication, with a threshold of viral load yet to be defined (given the frequent hematotoxicity of [val]ganciclovir and the frequent nephrotoxicity of foscarnet).

Secondly, the replication of *Herpesviridae* in the lungs of patients with severe respiratory failure of non-viral origin has been reported in various situations [30], including HSV in Covid-19 [31]. CMV and HSV detection was associated with longer mechanical ventilation (MV) in a study of patients with extracorporeal membrane oxygenation [32]; CMV detection was linked to the duration of MV, length of ICU stay, and 90-day all-cause mortality rate in immunocompetent patients with MV for various conditions [33]. CMV in BAL was even reported to be associated with longer ICU stay in patients with sepsis [34] even without lung failure. Taken together, these data could lead to the interpretation that CMV replication in lung is a bystander event (that is even more frequent in immunocompromised patients) and may reflect the severity of lung damage without necessarily contributing to them, or simply reflect the depth of immunocompromised state. In this interpretation, the association between disease severity and CMV viral load would be due to the fact that more immunocompromised patients may have the worst outcome of PJP and an increased risk of CMV replication, and that CMV infection *per se* does not impact PJP prognosis. However, the fact that CMV detection was associated with ICU admission independently of immune status (i.e., gammaglobulin blood concentration, total blood lymphocyte count) goes against this interpretation.

To clarify the interpretation of CMV detection in patients with PJP, the impact of CMV therapy should be studied. A clinical trial is nevertheless unlikely to be conducted, as it would only apply to low viral loads: indeed, it would be inappropriate to have an arm without antiviral therapy in patients with high viral loads. Retrospective studies are inconclusive on this point.

Interestingly, CMV detection has been associated with other invasive fungal infections (IFI): for example, a study [35] reported that haematopoietic stem cell transplantation patients with CMV diseases had a higher risk of *Aspergillus* invasive infection. The contribution of CMV to the physiopathology of IFI may thus be evoked. More simply, the neutropenia frequently associated with anti-CMV therapy may play an important role in these IFI.

Our study has several limitations, the first being its retrospective design. Among others, the diagnosis of PJP was not retrospectively reassessed (in particular, we did not reviewed the chest CT scan), and some patients we included may therefore have *P. jirovecii* colonisation rather than PJP, which may explain the relatively low mortality rate we observed among those admitted in ICU; however, the fact that the association between ICU admission and CMV detection was still significant when considering only patients with a high fungal load supports the potential link between CMV detection and PJP severity. In addition, the pragmatic proxy that we retained for severity, ICU admission, may be imperfect, as it may be influenced by bed availability, and limitation of life-support techniques (due to age or comorbidities); results could have been different if we had considered oxygen needs (maximal flow required for a SaO2>95%), or the extension of ground-glass opacity on CT scan, or the SaO2 or the PaO2 in room air. Another limit is that we did not expressed the *P. jirovecii* PCR results as a ratio of a household gene amplification, and these PCR values are therefore highly dependent from BAL dilution. Moreover, we did not assess the influence of antiviral therapy, which may have helped in interpretating the association between CMV detection and PJP outcome. However, it is likely that biases would have prevented us from drawing firm conclusions on this point: as stated, no specific antiviral guidelines exist regarding this situation, and antivirals may have been applied very heterogeneously. In addition, we were not able to totally exclude the

possibility that some patients may have had CMV prophylaxis when PJP was diagnosed. Moreover, we did not analysed CMV serology before the onset of PJP; this would had allowed us to discriminate primary infection from reactivation, which differs on several points, including the higher risk of CMV disease of the first. Lastly, some patients may have had pre-existent severe lung impairment (e.g., lymphoid interstitial pneumonia), which may have enhanced the risk of ICU admission.

## Conclusion

Blood and BAL detection of CMV is observed in one fourth of patients with PJP, and is associated with a higher rate of ICU admission in this population. Our data may lead to consider that CMV infection is detrimental *per se*, and not solely a reflection of inflammation and/or failing immune functions. This may suggest that antiviral CMV should be considered in patients with PJP and with CMV detection in blood or BAL, although a viral load threshold is still to be defined.

## Author Contributions

**Conceptualization:** Alexandre Perret, Marion Le Marechal, Raphaele Germi, Johan Noble, Aude Boignard, Olivier Epaulard.

**Data curation:** Alexandre Perret, Cécile Garnaud, Olivier Epaulard.

**Formal analysis:** Alexandre Perret, Raphaele Germi, Olivier Epaulard.

**Investigation:** Alexandre Perret, Olivier Epaulard.

**Methodology:** Olivier Epaulard.

**Project administration:** Olivier Epaulard.

**Resources:** Daniele Maubon, Cécile Garnaud, Johan Noble, Aude Boignard, Loïc Falque, Mathieu Meunier, Théophile Gerster.

**Validation:** Raphaele Germi, Daniele Maubon, Olivier Epaulard.

**Writing – original draft:** Alexandre Perret, Olivier Epaulard.

**Writing – review & editing:** Marion Le Marechal, Raphaele Germi, Daniele Maubon, Cécile Garnaud, Johan Noble, Aude Boignard, Loïc Falque, Mathieu Meunier, Théophile Gerster, Olivier Epaulard.

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
