## [Decision Letter · Decision Letter 0]

9 Oct 2023

PONE-D-23-27991Cytomegalovirus detection is associated with ICU admission in non-AIDS and AIDS patients with Pneumocystis jirovecii pneumoniaPLOS ONE

Dear Dr. Epaulard,

Thank you for submitting your manuscript to PLOS ONE. After careful consideration, we feel that it has merit but does not fully meet PLOS ONE’s publication criteria as it currently stands. Therefore, we invite you to submit a revised version of the manuscript that addresses the points raised during the review process.

We look forward to receiving your revised manuscript.

Kind regards,

Benjamin M. Liu, MBBS, PhD, D(ABMM), MB(ASCP)

Academic Editor

PLOS ONE

Journal Requirements:

Reviewers' comments:

Reviewer's Responses to Questions

**Comments to the Author**

1. Is the manuscript technically sound, and do the data support the conclusions?

Reviewer #1: Partly

Reviewer #2: Yes

2. Has the statistical analysis been performed appropriately and rigorously? 

Reviewer #1: No

Reviewer #2: Yes

3. Have the authors made all data underlying the findings in their manuscript fully available?

Reviewer #1: No

Reviewer #2: Yes

4. Is the manuscript presented in an intelligible fashion and written in standard English?

Reviewer #1: Yes

Reviewer #2: Yes

5. Review Comments to the Author

Reviewer #1: Since your aim is to use ICU admission to assess severity, therefore, 36.5% ICU admission is not enough to make a conclusion that “Cytomegalovirus detection is associated with ICU admission in non-AIDS and AIDS patients with Pneumocystis jirovecii pneumonia”. Kindly re-cast your topic to suite your main findings. Line 96-103 should be written in prose format. Also, no ethical approval obtained for the study.

Provide p value where you compare AID and Non-AID patients in table 1

105: the name of those genes should be specified

133: specify those continuous and categorical variables

134: Check for correct spelling of Chi square

222: Observed association with no p value

The whole conclusion should be re-casted to suite the original findings.

Reviewer #2: 1. In the introduction/discussion, the authors are encouraged to discuss the clinical significance, molecular targets and methodology of detection of Pneumocystis jirovecii in BAL. The authors should cite more references on this perspective, including the three references shown below. The detection of Pneumocystis jirovecii by PCR may be just colonization, rather than true infection. Did the patients under study have any imaging data to be included in this study?

McTaggart LR, Wengenack NL, Richardson SE. Validation of the MycAssay Pneumocystis kit for detection of Pneumocystis jirovecii in bronchoalveolar lavage specimens by comparison to a laboratory standard of direct immunofluorescence microscopy, real-time PCR, or conventional PCR. J Clin Microbiol. 2012 Jun;50(6):1856-9. doi: 10.1128/JCM.05880-11. Epub 2012 Mar 14. PMID: 22422855; PMCID: PMC3372158.

Alanio A, Desoubeaux G, Sarfati C, Hamane S, Bergeron A, Azoulay E, Molina JM, Derouin F, Menotti J. Real-time PCR assay-based strategy for differentiation between active Pneumocystis jirovecii pneumonia and colonization in immunocompromised patients. Clin Microbiol Infect. 2011 Oct;17(10):1531-7. doi: 10.1111/j.1469-0691.2010.03400.x. Epub 2011 Apr 12. PMID: 20946413.

Liu B, Totten M, Nematollahi S, Datta K, Memon W, Marimuthu S, Wolf LA, Carroll KC, Zhang SX. Development and Evaluation of a Fully Automated Molecular Assay Targeting the Mitochondrial Small Subunit rRNA Gene for the Detection of Pneumocystis jirovecii in Bronchoalveolar Lavage Fluid Specimens. J Mol Diagn. 2020 Dec;22(12):1482-1493. doi: 10.1016/j.jmoldx.2020.10.003. Epub 2020 Oct 15.

2. There is a disconnect between the objectives and the patients enrolled. The authors do not have to emphasize ICU patients in their goals. Though PJP severity was assessed by the requirement for intensive care unit (ICU) admission, there are different other metrics to evaluate the PCP severity.

3. Though the authors proposed that CMV viral load may be associated with PCP, there may be some confounding factors which may be the true reason of the co-incidence of high CMV and PCP. The immunocompromising status may be one of the true reasons. Please discuss this.

6. PLOS authors have the option to publish the peer review history of their article (what does this mean?). If published, this will include your full peer review and any attached files.

Reviewer #1: **Yes: **Pelumi Daniel Adewole

Reviewer #2: No

---

## [Author Response · Author response to Decision Letter 0]

22 Nov 2023

PONE-D-23-27991

Cytomegalovirus detection is associated with ICU admission in non-AIDS and AIDS patients with Pneumocystis jirovecii pneumonia

Answers to the comments of reviewers 

In your Data Availability statement, you have not specified where the minimal data set underlying the results described in your manuscript can be found. 

Upon re-submitting your revised manuscript, please upload your study’s minimal underlying data set as either Supporting Information files or to a stable, public repository and include the relevant URLs, DOIs, or accession numbers within your revised cover letter. 

The data can be find on the OSF depository, at the following link: https://osf.io/3kesf/files/osfstorage/6540f6be87852d1a56a5971c

Please include your full ethics statement in the ‘Methods’ section of your manuscript file. In your statement, please include the full name of the IRB or ethics committee who approved or waived your study, as well as whether or not you obtained informed written or verbal consent. If consent was waived for your study, please include this information in your statement as well. 

All patients had signed a consent form for the anonymous use of their medical data for research purposes (a decision they could revoke at any moment), and information on the retrospective use of anonymised data for research purpose was available for all patients. Due to this process, in accordance with French legislation, it was not required to submit this specific project to an ethical board. We clarified this in the “Methods” section in the revised version of the manuscript.

Reviewer #1

Since your aim is to use ICU admission to assess severity, therefore, 36.5% ICU admission is not enough to make a conclusion that “Cytomegalovirus detection is associated with ICU admission in non-AIDS and AIDS patients with Pneumocystis jirovecii pneumonia”. 

Having more than one third of the patient with the main outcome could be considered sufficient, in terms of statistical power: having 91 patients out of 249 allows performing robust, classical tests to explore the association of variables with this outcome.

Moreover, it is a rather consensual severity marker: it is likely that most patients admitted in ICU had severe Pneumocystis jirovecii pneumonia, and that most patients not admitted in ICU did not.

Kindly re-cast your topic to suite your main findings. 

We are not sure we fully understand this suggestion. Our main findings are about the association between CMV viral load and ICU admission. We rephrased the end of the introduction for better clarity in the revised version of the manuscript.

Line 96-103 should be written in prose format. 

This has been done in the revised version of the manuscript.

Also, no ethical approval obtained for the study.

All patients had signed a consent form for the anonymous use of their medical data for research purposes (a decision they could revoke at any moment), and information on the retrospective use of anonymised data for research purpose was available for all patients. Due to this process, in accordance with French legislation, it was not required to submit this specific project to an ethical board. We clarified this in the “Methods” section in the revised version of the manuscript.

Provide p value where you compare AID and Non-AID patients in table 1

There is no comparison between AIDS and not-AIDS patients in the manuscript.

105: the name of those genes should be specified

We used the gene of the major surface glycoprotein (this was already specified in the initial version of the manuscript).

133: specify those continuous and categorical variables

This would make the text maybe too cumbersome: the classification of variables is usually consensual, and in addition is usually obvious when reading tables.

134: Check for correct spelling of Chi square

This modification has been done in the revised version of the manuscript.

222: Observed association with no p value

We added some words in the revised version of the manuscript for more accuracy.

The whole conclusion should be re-casted to suite the original findings.

We modified the conclusion accordingly in the revised version of the manuscript. 

Reviewer #2 

In the introduction/discussion, the authors are encouraged to discuss the clinical significance, molecular targets and methodology of detection of Pneumocystis jirovecii in BAL. The authors should cite more references on this perspective, including the three references shown below. 

The 3 references may not be in the scope of our study: our aim was not to determine whether PCR on BAL was reliable to diagnose Pneumocystis jirovecii pneumonia, as this is something widely proven, but to study the association between CMV detection and the severity of Pneumocystis jirovecii pneumonia. However, we added these 3 references to the revised version of the manuscript.

The detection of Pneumocystis jirovecii by PCR may be just colonization, rather than true infection. 

This is an extremely relevant remark. We had acknowledged this is the initial version of the “Discussion” section, with the sentence “(…) the diagnosis of PJP was not retrospectively reassessed, and some patients we included may therefore have P. jirovecii colonisation rather than PJP”. 

Did the patients under study have any imaging data to be included in this study?

All the patients had a CT scan to retain the diagnosis of Pneumocystis jirovecii pneumonia. We choose not to include these data in the study, as it was not in its scope (even if another, different study considering, for example, the extension of Ground-glass opacity would also be interesting). We added a sentence on this point in the Discussion section of the revised manuscript.

There is a disconnect between the objectives and the patients enrolled. 

We notified 2 goals at the end of the introduction:

- to assess the frequency and intensity of CMV detection in non-AIDS and AIDS patients with Pneumocystis infection, 

- to assess the association between CMV detection and PJP severity.

To fulfill these goals, we included patients with a Pneumocystis jirovecii pneumonia who also had a test to detect CMV in blood or BAL.

Therefore, we do not identify any disconnection between these goals and the population we included.

The authors do not have to emphasize ICU patients in their goals.

There is no such emphasis on ICU admission; we did not mention it at the end of the introduction, where we specified the goals of the study; of course, ICU admission was not an inclusion criteria.

At the beginning of the Methods section, we wrote “Our main objective was to explore the association between outcome and CMV detection in patients with PJP” and then “The primary endpoint was the need for intensive care unit (ICU) admission related to the PJP episode”. As we chose ICU admission to determine PJP severity, it would have been complex not to specify it there.

Though PJP severity was assessed by the requirement for intensive care unit (ICU) admission, there are different other metrics to evaluate the PCP severity.

This is an interesting remark. The manuscript does not state that there are no other metrics to evaluate the PCP severity. We could indeed have used the need for oxygen (maximal flow required for a SaO2>95%), or the extension of ground-glass opacity on CT scan, or the SaO2 or the PaO2 in room air. 

The definition for severity we retained is more pragmatic, and may reflect more the concerns of clinicians.

We acknowledged the fact that other metrics could have been used for severity in the revised version of the manuscript. 

3. Though the authors proposed that CMV viral load may be associated with PCP, there may be some confounding factors which may be the true reason of the co-incidence of high CMV and PCP. 

This is a relevant remark. In the submitted version of the manuscript, we had written “Taken together, these data could lead to the interpretation that CMV replication in lung is a bystander event (that is even more frequent in immunocompromised patients) and may reflect the severity of lung damage without necessarily contributing to them, or simply reflect the depth of immunocompromised state.”

The immunocompromising status may be one of the true reasons. Please discuss this.

This is an important point. In the submitted version of the manuscript, we had written “(…) the association between disease severity and CMV viral load would be due to the fact that more immunocompromised patients may have the worst outcome of PJP and an increased risk of CMV replication, and that CMV infection per se does not impact PJP prognosis.”

---

## [Editor Report · Decision Letter 1]

18 Dec 2023

Cytomegalovirus detection is associated with ICU admission in non-AIDS and AIDS patients with Pneumocystis jirovecii pneumonia

PONE-D-23-27991R1

Dear Prof Epaulard,

We’re pleased to inform you that your manuscript has been judged scientifically suitable for publication and will be formally accepted for publication once it meets all outstanding technical requirements.

Kind regards,

Benjamin M. Liu, MBBS, PhD, D(ABMM), MB(ASCP)

Academic Editor

PLOS ONE

Additional Editor Comments (optional): Please note that I have acted as a reviewer for this manuscript, and you will find my comments below, under Reviewer 2.
---

## [Editor Report · Acceptance letter]

2 Jan 2024

PONE-D-23-27991R1 

PLOS ONE

Dear Dr. Epaulard, 

I'm pleased to inform you that your manuscript has been deemed suitable for publication in PLOS ONE. Congratulations! Your manuscript is now being handed over to our production team.

Kind regards, 

on behalf of

Dr. Benjamin M. Liu 

Academic Editor

PLOS ONE